# A Novel Approach in Skin Care: By-Product Extracts as Natural UV Filters and an Alternative to Synthetic Ones

**DOI:** 10.3390/molecules28052037

**Published:** 2023-02-21

**Authors:** Sara M. Ferreira, Sandra M. Gomes, Lúcia Santos

**Affiliations:** 1LEPABE—Laboratory for Process Engineering, Environment, Biotechnology and Energy, Faculty of Engineering, University of Porto, Rua Dr. Roberto Frias, 4200-465 Porto, Portugal; 2ALiCE—Associate Laboratory in Chemical Engineering, Faculty of Engineering, University of Porto, Rua Dr. Roberto Frias, 4200-465 Porto, Portugal

**Keywords:** bioactive compounds, natural ingredients, antioxidants, UV filters, sun protection factor, onion peels, passion fruit peels

## Abstract

The cosmetic industry has been focusing on replacing synthetic ingredients with natural ones, taking advantage of their bioactive compounds. This work assessed the biological properties of onion peel (OP) and passion fruit peel (PFP) extracts in topical formulations as an alternative to synthetic antioxidants and UV filters. The extracts were characterized regarding their antioxidant capacity, antibacterial capacity and sun protection factor (SPF) value. Results revealed that the OP extract exhibited better results, which can result from the high concentrations of quercetin, as identified and quantified in HPLC analysis. Afterward, nine formulations of O/W creams were produced with minor changes in the quantity of additives: OP and PFP extract (natural antioxidants and UV filters), BHT (synthetic antioxidant) and oxybenzone (synthetic UV filter). The stability of the formulations was determined for 28 days; it was verified that they remained stable throughout the study period. The assays of the formulations’ antioxidant capacity and SPF value revealed that OP and PFP extracts have some photoprotective properties and are excellent sources of antioxidants. As a result, they can be incorporated in daily moisturizers with SPF and sunscreens replacing and/or diminishing the quantities of synthetic ingredients, reducing their negative effects on human health and the environment.

## 1. Introduction

The history and trends in beauty standards have always been shaped in parallel with the evolution of humanity. Indeed, the cosmetic market has transformed drastically throughout the decades, passing through the reliance on fishing, hunting and superstitions in its early days, turning towards medicine and pharmacies, and now focusing on natural and green materials [1]. Scientific research brought to light the negative effects of synthetic ingredients used in cosmetic formulations and consumers’ concern regarding this topic, which led them to be more refined and aware of the cosmetics they buy. Consequently, the cosmetic industry has been focusing on developing green or natural products, made of ingredients of natural origin. The production of green cosmetics is in line with ecological standards, seeking to reduce water, material and energy consumption, reduce pollution and develop recycled packaging. Furthermore, green cosmetics also contain bioactive ingredients, from natural sources, that have beneficial effects on the consumer [2]. The most recent trend in the cosmetic field is focused on the research of natural ingredients that delay skin aging.

The concept of wellness and well-being has changed over the past years; it includes beauty, health, fitness and antiaging treatments. Indeed, today’s consumers express awareness regarding their skin wellness and sun protection, and studies have shown that age-prevention treatments are one of the main prioritized categories [3,4]. Photoprotection is one of the main used techniques to delay skin aging and guarantee skin well-being. This term refers to the preventative and therapeutic measures employed to protect human skin against skin cancer (photocarcinogenesis) and skin aging (photoaging) derived from UVB and UVA radiation exposure. Photoprotective measures include applying broad-spectrum sunscreens before sun exposure, wearing protective clothing and accessories such as sunglasses and hats and avoiding sun exposure [5,6]. Sunscreen is a cosmetic product capable of protecting the human skin from ultraviolet radiation due to the presence of molecules, UV filters that may be physical, chemical or both, that are capable of absorbing, dispersing or reflecting UVB and UVA radiation [7]. Sunscreen formulations can be oil-in-water (O/W) or water-in-oil (W/O) emulsions; however, consumers prefer O/W formulations because they are easy to use, dry quickly and are non-occlusive. Additionally, organic filters and inorganics may be present. Along with finding the appropriate formula, selecting products with suitable ingredients can significantly increase their effectiveness. Consumers demand an all-in-one sunscreen product that is non-toxic, non-allergic, moisturizing, antioxidant and protects against both UVA and UVB rays with a high sun protection factor (SPF) number. Figure 1 depicts the typical components of an O/W sunscreen formulation, along with their functions [8].

UV filters are a key ingredient in sunscreen formulations and can be classified into two groups according to their nature and mechanism of action: inorganic filters and organic filters. Inorganic filters, such as zinc oxide (ZnO) and titanium dioxide (TiO_2_), reflect and scatter light, offering broad-spectrum protection from UVA and UVB rays, which makes them broad-spectrum filters. They are soft enough for daily use because they rarely irritate the skin and are being advertised as safer alternatives to organic filters [9]. However, due to the scattering effect, they frequently generate a so-called “whitening” effect when applied to the skin, which has a detrimental effect on the aesthetics and efficacy of sunscreen lotions. Although inorganic filters are considered a safer option than organic ones, there are some disadvantages. Studies showed that there is evidence that UV irradiation of metal oxides in these products can result in hazards, confirming that the generation of reactive oxygen species (ROS) and degradation of organic compounds can occur and that there are no data to support their long-term safety [10,11]. Organic UV filters are compounds used to absorb UVA and/or UVB radiation, transforming it into energetic radiation with a wavelength superior to the one of the incident radiation. The photochemical stability of the molecule is an important factor since it must be able to repeat the absorption process multiple times without degrading [5,12]. When compared with inorganic filters, these agents display terrific properties, including stability and being nonirritant, nonvolatile, non-photosensitizing and non-staining on human skin [13]. The majority of the organic UV filters are not able to absorb radiation in the entire UV spectrum; depending on the wavelength range they can absorb, they can be divided into UVA filters (from 315 to 400 nm), UVB filters (from 280 to 315 nm) and broad-spectrum filters (UVA and UVB) [5]. To achieve effective sun protection, each sunscreen must have the concentration of active substances adjusted; furthermore, due to the synergy between the active ingredients, a combination of organic and inorganic filters should be used since it allows balanced sun protection. Figure 2 displays the main differences between chemical and physical sunscreens and the main used UV filters in each case.

One of the main challenges in sunscreen formulation is the production of photostable cosmetics that maintain an unaltered appearance and function when exposed to UV radiation [5]. Indeed, with exposure to sunlight, many sunscreen ingredients degrade and lose their photoprotective qualities, reducing the product’s efficacy. Additionally, sunscreens must remain on the epidermis’s surface to ensure efficacy and provide the best protection against UVA/UVB rays [14]. The cutaneous system communicates with other systems or organs of the human body, such as the brain and endocrine organs. Therefore, when UV radiation contacts the skin, changes can occur not only locally but also in other parts of the body. For example, UV radiation can stimulate the hypothalamic–pituitary–adrenal system, regulating the expression of different molecules or receptors [15]. Excessive exposure to radiation can intensify the upregulation or downregulation of those molecules, disrupting systemic homeostasis. It is also important that UV filters stay on the surface of the skin since recent research has revealed that when specific certain UV filters are absorbed through the skin, their effectiveness and safety are affected, posing a risk to consumers once they reach systemic circulation [16,17]. This absorbance may have various detrimental effects including the induction of apoptosis and a rise in mutagenesis or estrogenic activity, resulting in the rapid proliferation of cancer cells [17,18,19,20]. Furthermore, UV filters are becoming emergent contaminants due to their omnipresent occurrence in the environment. These molecules can easily enter aquatic systems, either directly or through the washing of the skin, and some studies reported their presence in different environmental compartments such as surface waters, oceans, seas, coastal waters, groundwater, lakes, rivers and sediments [12,20,21].

Botanical agents—compounds obtained from herbs, stems, peels, roots and other elements of plant origin—are gaining attention and becoming more popular as an alternative to typical synthetic UV filters. These agents are secondary metabolites produced by different plants and are crucial for their growth and continuity. The literature reports that these metabolites possess a wide array of biological properties, such as antioxidant, antibacterial and anti-inflammatory properties, as well as the capacity to absorb UV radiation. The most common botanicals that display the previous properties are vitamin C, vitamin E and plant extracts, rich in phenolic compounds (PCs) [11]. Considering PCs, their capacity to absorb UV rays stems from the presence of aromatic rings containing electron resonances [9]. Indeed, UV filters that absorb UV radiation usually possess aromatic rings in their structure with a carbonyl group; they receive the photons from the UV radiation and undergo conformational changes in their molecular structure, release the incident energy as heat or emit radiation as a higher wavelength [22]. Resveratrol and quercetin are commonly known for their antiaging properties; indeed, these molecules prevent DNA damage since they decrease the formation of free radicals. Additionally, the literature reveals that these molecules display a high capacity to absorb UV radiation and, consequently, can be used as organic UV filters [23].

Phenolic compounds are the main secondary metabolites of plants; they are responsible for astringency and pigmentation, acting also as protective agents against UV light, parasites and insects. These compounds can be found in fruits and vegetables and are present in both edible and non-edible plant parts (for example, roots, stems and seeds) [24,25]. An interesting approach is to obtain PCs from by-products of the agri-food industry, which are generated in large amounts, are considered an environmental problem and have no economic value. However, the literature reveals that these by-products are rich in bioactive compounds, such as PCs, making them a cheap natural source for obtaining them.

Literature studies revealed that onion peels (OPs) and passion fruit peels (PFPs) are rich sources of bioactive phenolic compounds, such as polyphenols (for example, resveratrol), flavonoids (such as kaempferol and quercetin) and phenolic acids (such as gallic acids) [26,27]. Furthermore, an advantage of using these by-products is the ease of obtaining them, since the peels are easy to separate and most restaurants and industries discard them. Therefore, due to their bioactive potential, OP and PFP are interesting and attractive sources for obtaining extracts rich in phenolic compounds to incorporate into cosmetic products. Furthermore, this strategy promotes the reuse of agro-industrial by-products and the valorization of their bioactive composition, and it allows reducing the environmental problems caused by their incorrect treatment and disposal.

The present work intended to assess the potential of phenolic extracts, from agro-industrial by-products, to reduce the quantity and/or replace synthetic UV filters in a sunscreen formulation. Hence, OP and PFP were selected as by-products due to their high content of PCs and because they are easy to obtain and clean. Therefore, this study aimed to develop a value-added and sustainable face moisturizing cream with SPF through the incorporation of UV filters from natural sources, following the concept of the circular economy.

## 2. Results and Discussion

### 2.1. Characterization of the Phenolic Extracts

The present study intended to evaluate the potential of phenolic extracts, of by-products from the agri-food industry, to reduce the quantity and replace synthetic UV filters in creams with SPF capacity. Onion peel (OP) and passion fruit peel (PFP) were the by-products selected to obtain extracts rich in phenolic compounds (PCs). Afterward, the achieved extracts were characterized regarding their biological properties, mainly their total phenolic content (TPC); their antioxidant capacity, employing the assays with DPPH and ABTS; their antibacterial capacity; and lastly their protection against UV radiation through the determination of the sun protection factor (SPF). The obtained results are displayed in Table 1.

Considering the antibacterial capacity, it is observable that none of the extracts display activity against the Gram-negative bacteria in the study, *E. coli*. On the other hand, both extracts exhibited the capacity to inhibit the growth and development of both *S. aureus* and *S. epidermidis*, which are characterized as Gram-positive bacteria. These bacteria were chosen as model microorganisms since *E. coli* and *S. aureus* are pathogens commonly related to human infections and *S. epidermidis* is an opportunistic microorganism naturally found in the skin. Additionally, it is possible to observe that, for both extracts, an increase in the extract concentration contributed to an increment in the inhibitory effect towards the Gram-positive bacteria. The obtained results regarding the type of bacteria were expected. Indeed, the literature shows that phenolic extracts do not possess the capacity to inhibit the growth of Gram-negative bacteria [28]; this effect is associated with their cell structure which is constituted by a two-layer cell membrane, the outer part of which displays strong hydrophilicity, making these microorganisms less susceptible to the action of the extracts [29]. In addition, the literature also suggests that the diameter of the inhibition halos is directly proportional to the TPC; this relation is observed in the obtained results since the OP extract exhibited higher TPC and antibacterial capacity in comparison with the PFP extract.

Considering the total phenolic content analysis, it is possible to observe, from Table 1, that the obtained values were in the range of 124–377 mg_GAE_/mL. Furthermore, it is visible that the OP extract displayed a significantly higher content of phenolics than the PFP extract. Regarding the antioxidant analysis, the results of the assay with the DPPH radical demonstrated that the IC_50_ of the PFP extract is significantly higher than the one observed in OP; this result means that higher concentrations of PFP extract are needed to inhibit 50% of DPPH when compared to OP extract. Additionally, the results in Trolox equivalents, from both assays, reveal that the OP extract exhibited higher values and possessed higher antioxidant capacity than the PFP extract. Even though both extracts are rich in phenolic compounds, it is possible to conclude from the results that the OP extract displays higher antioxidant capacity, making it more interesting to be used in the cosmetic industry. The main reason for the different performances of the extracts can be associated with the composition of the extracts, for example, the main phenolic compounds present in the matrix and their concentration, since the molecular structure can influence the antioxidant capacity [30]. Therefore, a high-performance liquid chromatography with photodiode array detection (HPLC/DAD) analysis was performed to comprehend the difference between the extracts’ behaviors. The results are presented in Table 2.

From the results in Table 2, it is possible to observe that the OP extract displays a higher concentration of the analyzed phenolic compounds, apart from the gallic acid concentration, which is similar to the concentration in the PFP extract. The compound present in higher concentration in OP extract was quercetin (approximately 26 mg_quercetin_/g_extract_), followed by resveratrol. Indeed, it was expected that quercetin was the major phenolic compound present in this extract since onions are known to be good sources of quercetin; the literature reports that even with different extraction techniques, quercetin and its derivatives (such as quercetin glycosides) are usually the main compounds identified in OP extract [26]. The results of the HPLC analysis explain the differences between the extracts’ behaviors regarding their antioxidant capacity. Of the phenolic compounds, quercetin and resveratrol are known to prevent damage to DNA, since they can decrease the formation of free radicals, which results in the high antioxidant capacity associated with these compounds [23]. Therefore, the difference between the antioxidant capacity verified in OP and PFP extracts can be explained by the higher concentrations of quercetin and resveratrol present in OP extract.

The SPF value represents a relative measurement of the time a sunscreen or a cosmetic product will protect the skin from UV rays, protecting it from sunburns; an increase in the SPF value leads to an increase in sunburn protection. Therefore, the SPF was determined for both extracts in the study accordingly with Equation (3) presented in Section 3.2.9, and the results are displayed in Table 3.

Ultraviolet absorption peaks were observed for oxybenzone at 279 and 317 nm, OP extract at 288 and 365 nm and PFP extract at 281 and 367 nm. As shown in Table 3, the OP extract exhibited the highest SPF value. However, the concentration of oxybenzone was 20 times lower than the one used for the extracts to obtain an absorbance spectrum between 0 and 1. The results are following the literature since it is described that phenolic derivatives display a high capacity to absorb UV radiation, exhibiting SPF values that can range between 7 and 29 [23]. Nevertheless, the obtained results prove that phenolic extracts from OP and PFP represent a promising alternative as a UV filter in sunscreens. As previously seen, the most abundant phenolic present in the extracts is quercetin; however, it is present in different concentrations. The molecular structures of the analyzed phenolics are exhibited in Figure 3A.

The analyzed compounds belong to two different classes of phenolic compounds: non-flavonoids (gallic acid and resveratrol) and flavonoids (kaempferol and quercetin). The main difference between these two classes is the basic structure of the compounds. In the case of flavonoids, the general structure is composed of a 15-carbon skeleton, containing two benzene rings (A and B) linked via a heterocyclic pyran ring (C), as exhibited in Figure 3B. The different classes of flavonoids differ in the oxidation level and pattern of substitution of the C ring [31]. As seen before, chemical compounds that act as UV-absorbing agents usually have an aromatic ring and a carbonyl group in their structure. Since gallic acid and resveratrol do not possess a carbonyl group in their structure, it is unlikely that they hold the capacity to absorb UV radiation. On the other hand, as it is possible to observe from Figure 3, both quercetin and kaempferol have a double bond between carbons 2 and 3 on the C ring, conjugated with the 4-carbonyl group also on the C ring. This molecular structure gives these compounds an unsaturated heterocyclic C ring, which promotes a conjugation between the A and C rings; an increase in the conjugation leads to an increase in the absorption wavelength of the molecule, which allows them strongly absorb UV radiation [32,33,34]. Therefore, the higher concentrations of kaempferol and quercetin present in OP extract, when compared to the PFP extract, can be the cause for the higher SPF value observed for this extract.

Overall, the results prove that phenolic extracts from agro-industrial by-products have a vast set of biological properties, such as antioxidant and antibacterial capacity, as well as the capacity to absorb UV radiation, making them extremely interesting for applications in the cosmetic industry. However, although not included in the scope of the developed work, it is of great importance to study the potential adverse effects of OP and PFP on human health. Therefore, it would be of great interest to evaluate the toxicity and the photostability of the OP and PFP extracts. For example, the cytotoxicity of the extracts could be evaluated in keratinocytes, the most common cell type in the skin, since the final product is intended for topical applications. This test could give us an indication of which concentrations of OP and PFP extracts are safe for skin cells.

### 2.2. Evaluation of the Stability of the Sunscreens

The developed sunscreen formulations include only the essential ingredients to create a cream to analyze the effect of the antioxidant capacity and SPF after adding the extracts or synthetic compounds, with the same functions. The quantities (%) of all used ingredients are in accordance with those reported in the literature and follow the legislation. To achieve the goal of this work, nine oil-in-water (O/W) formulations were produced. Synthetic antioxidant (BHT) was used as a positive control since it was necessary to develop an experiment using a synthetic AO to compare it to the formulations containing the phenolic extracts. Additionally, oxybenzone, a commonly used synthetic UV filter in sunscreens, was also selected as a positive control to use as a comparison in the study of the SPF of the formulations containing the phenolic extracts. The formulations were named NC (Negative control) for the sample without additives, PC-AOx (positive antioxidant control) for the sample containing BHT, PC-UV (positive UV filter control) for the formulation with oxybenzone, OP-AOx (onion peel extract antioxidant sample), PFP-AOx (passion fruit peel extract antioxidant sample), Mix-AOx (mixture of antioxidant control and OP and PFP extract), OP-UV (onion peel extract UV filter sample), PFP-UV (passion fruit peel extract UV filter sample) and Mix-UV (mixture of UV filter control and OP and PFP extract). The produced formulations are presented in Figure 4.

After the formulations were manufactured, their stability was evaluated for 28 days, during which the formulations were subjected to various tests. Organoleptic qualities, such as color and smell, stayed relatively unaltered throughout the research. Formulations NC, PC-AOx and PC-UV displayed a white color; formulations OP-AOx, PFP-AOx, Mix-AOx and PFP-UV had a slightly darker color; and formulations OP-UV and Mix-UV had an old rose color due to the presence of the OP phenolic extract. In terms of fragrance, the formulations that contained the phenolic extracts exhibited a light aroma; nevertheless, no alteration of the smell was detected. Regarding appearance, all the formulations lacked visible particles, indicating that the compositions were homogeneous. It is noteworthy that the addition of the extracts and the increase in the concentration of the extracts in the formulations did not originate visible alterations in the viscosity and spreadability of the sunscreens.

The pH value of a cosmetic formulation is one of the main chemical properties that should be studied since changes in this value can provide information regarding product instability or the existence of any contamination. The pH results are displayed in Figure 5.

From the obtained results, it is possible to observe that the pH values were in the range of 4–6. Since the pH value of the human skin is within the values of 4.5–6, products intended for topical application should have a pH contained in this range [35]. Therefore, it is observable that the pH of the formulations is in accordance with the acceptable range for topical formulations, apart from that of sample MIX-UV which is slightly more acidic. Moreover, it is perceptible that the addition of the phenolic extracts from OP and PFP leads to a decrease in the pH value of the formulations, which follows the literature since phenolics and flavonoids display slightly acidic characteristics [36].

With the intent to evaluate the stability of the formulations under extreme conditions, accelerated stability assays were performed. After the centrifugation assay (a test that allows testing the physical stability of the formulations in extreme conditions), it was noticeable, as shown in Figure 6, that none of the formulations exhibited phase separation. Additionally, the formulations were subjected to a thermal stability test, a standard procedure for simulating long-term shelf life. The results indicated that no phase separation occurred during these stability tests and that there were no significant changes in viscosity, weight or color. As a result, it can be stated that the prepared emulsions exhibit exceptional long-term stability, while their properties remain unaffected by temperature changes.

The formulations were subjected to an antibacterial capacity assay, following a similar procedure to the one performed with the extracts, at t_0_ and t_2_. The obtained results are exhibited in Table 4.

From the results in Table 4, it is possible to conclude that no antibacterial capacity was detected against *E. coli*, which was not unexpected since both OP extract and PFP extract did not exhibit the capacity to inhibit the growth of this microorganism. Regarding *S. aureus*, it is possible to observe that all the formulations exhibited similar behavior in terms of inhibiting this bacterium; moreover, the storage time did not influence the antibacterial capacity of the formulations, proving that they can inhibit the growth of *S. aureus* over time. Considering *S. epidermidis*, the results demonstrate that there were some significant changes in the values between the formulations and over time; nevertheless, it is noticeable that the developed sunscreens can inhibit the growth of the studied microorganism. However, the results demonstrate that all the formulations have an analogous behavior to the formulation NC (that works as negative control), which may mean that the antibacterial capacity of the samples is not dependent on the addition of extracts, but on the composition itself. Indeed, the base formulation contains cocamidopropyl betaine, an amphoteric emulsifier, that is reported to exhibit antibacterial capacity, which can explain this biological property of the produced formulations [37]. The antibacterial capacity is an important property of the formulations since it allows for the maintenance of the integrity of the formulations.

Since moisturizing creams possess oils and fats in their formulation, which are materials extremely susceptible to oxidation, it is important to evaluate the extent of their oxidation. The most used chemical approach for assessing the oxidative degradation of oils is the use of peroxide values (PVs), which quantify the concentration of hydroperoxide. The redox interaction between hydroperoxides and an excess of KI in an acidic media leads to the stoichiometric release of molecular iodine, which is then titrated against a thiosulfate solution and is the basis for the PV measurements [38]. This assay allows determining the extent of the primary oxidation of the formulations, and it allows determining the oxidation state of oils and fats. The obtained results are presented in Figure 7.

The literature states that PVs up to 5 mEq/kg indicate as a low oxidation state, values between 5 and 10 mEq/kg indicate a moderate state of oxidation and values above 10 mEq/kg indicate a high oxidation state [39]. From the results in Figure 7, it is possible to observe that all the formulations display a low oxidation state and that formulation NC exhibited the highest PV value, while formulations Mix-AOx and Mix-UV demonstrated the lowest PVs. It was anticipated that formulation NC presented the highest PV due to it being the negative control and not having any compound with antioxidant capacity incorporated. The incorporation of the extracts in formulations OP-AOx and PFP-AOx allowed for a decrease in the PV when compared with the negative control; however, when compared with formulation PC-AOx (containing BHT), the PVs obtained for both formulations were superior. Nevertheless, it is possible to observe that the increase in the extracts’ concentration originates from a decrease in the extent of the oxidation state since it reduces the PV. This is an interesting result because BHT is a synthetic antioxidant and is limited to 0.1–0.5%; since phenolic extracts may contain additional beneficial chemical compounds, such as vitamins, they can be added in greater quantities than BHT, hence increasing the formulations’ resistance to oxidation. These results prove that phenolic extracts can help prevent the oxidation of a cosmetic product, giving it oxidative stability, and display similar results to synthetic antioxidant BHT when incorporated in higher concentrations. Even though the PVs of the formulations are low and decrease over time, this does not mean that there has been no oxidation or that the oxidation has stopped. As a result, to acquire more reliable and full information about the degree of lipid oxidation in the formulations, PV determination should be supplemented with secondary oxidation product analysis.

### 2.3. Determination of the Antioxidant Properties of the Formulations

In order to determine the potential of the phenolic extracts from OP and PFP to act as antioxidants in the developed formulations, the antioxidant capacity of the formulations was assessed. The determination of the antioxidant capacity was performed using the assays with the DPPH and ABTS radicals, and the results are expressed as a percentage of inhibition for each radical. The obtained values are displayed in Figure 8.

From the results displayed in Figure 8A,B, it is possible to observe that the antioxidant capacity of the formulations containing additives (BHT, OP extract and PFP extracts) exhibited a higher capacity to inhibit both DPPH and ABTS radicals than the negative control. Nevertheless, it is noticeable that there is a decrease in the scavenging capacity of the formulations over time; these results follow the literature reports [40,41]. Indeed, antioxidants are biological species that are capable of reducing other molecules and can capture free radicals, stabilizing them; however, the properties displayed by these molecules are also one of their limitations, as they tend to be easily degraded and oxidized. Therefore, the reduction in the antioxidant capacity of the formulations can be associated with the degradation of the antioxidant compounds present in the composition. Comparing formulations OP-AOx and PFP-AOx with the positive control (PC-AOx), it is possible to observe that the formulations containing phenolic extract display a similar behavior regarding the inhibition of the radicals in the study. Additionally, from Figure 8A, it appears that over time the OP-AOx and PFP-AOx formulations maintain their ability to inhibit relatively high percentages of DPPH compared to the PC-AOx formulation, whose antioxidant capacity decreases considerably. Lastly, focusing on formulation Mix-AOx, it is possible to observe that this sample exhibited similar results to the other samples (except formulation NC) in the assay with the ABTS radical. Regarding the assay with the DPPH radical, this sample exhibited the best performance throughout the study since it was able to inhibit higher percentages of this radical. Therefore, the obtained results indicate that it is possible to incorporate phenolic extracts from by-products into cosmetic formulations as a replacement and/or to diminish the quantity of synthetic antioxidants, such as BHT, used.

### 2.4. Sun Protection Factor of the Sunscreens

In order to determine the efficacy of the prepared sunscreen emulsions, regarding UV radiation protection properties, the sun protection factor (SPF) was determined. It is noteworthy that for this assay only formulations NC, PC-UV, OP-UV, PFP-UV and Mix-UV were analyzed since they were the ones to which UV filters were added. To calculate the SPF values, the absorbance values at specific wavelengths were required; the absorption spectra of these formulations are shown in Figure 9. The SPF values were determined using Equation (3) presented in Section 3.2.9. The obtained values are listed in Table 5.

From Table 5, it is possible to observe that formulation NC has the lowest SPF values, which is compatible with the fact of no UV filter was added since this was the negative control. Regarding the formulation PFP-UV, the SPF value was considerably lower than the ones for PC-UV and OP-UV. A probable explanation for this can be a reduction in the photoprotective capacity of the PFP extract when incorporated into a cosmetic product. Formulation PC-UV with 5% oxybenzone displayed an SPF value of 10.10, which is in accordance with literature values, and this value was higher than that of the formulation OP-UV with 5% OP extract. The formulation Mix-UV, containing 1.6% of oxybenzone and 1.6% of each extract (OP and PFP), exhibited a higher SPF value when compared with formulations OP-UV and PFP-UV. This result can suggest a synergistic effect when phenolic extracts are added to a formulation with oxybenzone. Synergistic effects between plant extracts and synthetic UV filters have been documented in the literature, where the addition of a plant extract containing phenolic compounds increases the SPF value of sunscreen [42]. Even though the SPF value of formulation Mix-UV was lower than the one displayed by formulation PC-UV, it is important to consider that the concentration of the synthetic UV filter was considerably reduced. This result shows that the incorporation of phenolic extracts from OP and PFP allows diminishing the concentration of synthetic UV filter in the formulations produced, allowing interesting SPF values to be reached. Furthermore, as previously stated, phenolic extracts contain a mixture of bioactive compounds which allows them to be added in greater quantities than oxybenzone, which can lead to an increase of the SPF value and subsequently to an upsurge in the protection against UV radiation. Therefore, the obtained results show that phenolic extracts from agro-industrial by-products possess some photoprotective properties and could be used in topical formulations.

Even though the developed research exhibited promising results regarding the incorporation of phenolic extracts from agro-industrial by-products into sunscreens, further studies regarding the stability of the formulation should be performed. Since ultraviolet radiation can reduce the photoprotective ability of cosmetics and can also produce free radicals, photostability is another crucial factor determining the efficacy and safety of sunscreen products. However, the degree of degradation depends on the molecular structure of the constituents. Therefore, the photostability of the formulations and the extracts should be assessed, in future work, to ensure the safety of the use of sunscreen formulations.

## 3. Materials and Methods

### 3.1. Samples and Reagents

The onion peels were collected from a local supermarket in Vila Nova de Gaia, Portugal, and were identified as being originated from Spain. The passion fruit peels were collected from Arentim, Braga, Portugal. The extraction solvent ethanol (Ref. 1.02371.1000, C_2_H_6_O, CAS 64-17-5) was obtained from VWR (Rosny-sous-Bois, France). For the antioxidant and antimicrobial capacity assays, 2,2-diphenyl-1-picrylhydrazyl (DPPH) (Ref. D9132, C_18_H_12_N_5_O_6_, CAS 1898-66-4) and ascorbic acid (AA) (Ref. A5960, C_6_H_8_O_6_, CAS 50-81-7) were used and purchased from Sigma Aldrich (St. Louis, MO, USA). For the HPLC analysis, the standards gallic acid (Ref. 147915, C_7_H_6_O_5_, CAS 149-91-7), kaempferol (Ref. 60010, C_15_H_10_O_6_, CAS 520-18-3), quercetin (Ref. Q4951, C_15_H_10_O_7_, CAS 117-39-5) and resveratrol (Ref. R5010, C_14_H_12_O_3_, CAS 501-36-0) were purchased from Sigma Aldrich (St. Louis, MO, USA), and the solvents used were ethanol, ultrapure water and acetonitrile (Ref. 83639.320, C_2_H_3_N, CAS 75-05-8) purchased from Sigma Aldrich (St. Louis, MO, USA). For the lipid oxidation assay, barium chloride dihydrate (Ref. 217565, BaCl_2._2H_2_O, CAS 10326-27-9) and iron chloride III (ref. F2877, FeCl_3._7H_2_O, CAS 10025-77-1) were obtained from Sigma Aldrich (St. Louis, MO, USA), iron sulfate (II) heptahydrate (Ref. 24244.232, FeSO_4_∙7H_2_O, CAS 7782-63-0) and hydrochloric acid (Ref. 20255.290, HCl, CAS 7647-01-0) were bought from VWR (Rosny-sous-Bois, France), ammonium thiocyanate (Ref. A10632, CH_4_N_2_S, CAS 1762-95-4) was acquired from Alfa Aesar (Haverhill, MA, USA) and chloroform (Ref. 438607, CH_3_Cl, CAS 67-66-3) and methanol (Ref. 414816, CH_3_OH, CAS 67-56-1) were purchased from Carlo Erba (Barcelona, Spain). Merck Millipore Mill-Q water purification equipment, with 18.2 Ω of electric resistance (Billerica, MA, USA), was used for deionized water.

### 3.2. Methods

#### 3.2.1. Extraction of Bioactive Compounds

The onion peels (OPs) and passion fruit peels (PFPs) were washed, and the excess water was removed using a paper towel. Thenceforth, the peels were dried at 50 °C in a drying oven (BINDER GmbH, Tuttlingen, Germany), milled with a coffee grinder (Q.5321 Qilive, Auchan, Croix, France) to obtain particles below 18 mesh (1 mm) and stored in a desiccator at room temperature, protected from light. The bioactive compounds were extracted from both OP and PFP using a solid–liquid extraction with a Soxhlet apparatus. The extraction procedure was adapted from the literature [37]. Ethanol was used as a solvent, with a ratio of 1:20 m/V. The extractions were conducted for 1.5 h, in triplicate. Then, the solvent was evaporated using a rotary evaporator (Rotavapor R-200, BUCHI Laboratories, Flawil, Switzerland), followed by a gentle stream of nitrogen.

#### 3.2.2. Phenolic Compound Quantification

High-performance liquid chromatography (HPLC-DAD) was used to identify and quantify the phenolic compounds (PCs) present in the extracts and was performed using an Elite LaChrom (Hitachi, Japan) HPLC system, equipped with a Hitachi L-2200 autosampler, L-2130 pump and L-2455 diode array detector. The samples were dissolved in acetonitrile:water:ethanol (2:1:1 *v*/*v*/*v*) and injected in a Puroshper STAR RP-18 LiChroCART (250 mm × 4.0 mm, 5.0 μm) chromatography column (Merck, Germany). The mobile phases A and B were Milli-Q water with 0.5% of orthophosphoric acid and methanol:acetonitrile (80:20 *v*/*v*), respectively. The PCs were identified by the retention time (RT) and quantified using the external standard method at 280 nm for gallic acid, 305 nm for resveratrol and 365 nm for kaempferol and quercetin.

#### 3.2.3. Characterization of the Phenolic Extracts

The antioxidant properties of the OP and PFP extracts were analyzed by the determination of the total phenolic content (TPC), and the radical scavenging activity was evaluated against 2,2-diphenyl-1-picrylhydrazyl (DPPH) and 2,2′-azino-bis(3-ethylbenzothiazoline-6-sulfonic acid) (ABTS) radicals.

The TPC was determined using the Folin–Ciocalteu method, following a literature protocol with adaptation [43]. The OP and PFP extracts were dissolved in ethanol at a concentration of 1000 mg/L. Briefly, 20 μL of the sample, 100 μL of Folin–Ciocalteu phenol reagent and 1580 μL of water were added to a 2 mL cuvette. Afterward, 300 μL of sodium carbonate solution (333.3 g/L in water) was added. The cuvettes were left to incubate in the dark for 2 h, at room temperature. Finally, the absorbance was determined at 750 nm using a Thermo GENESYS 10S UV-Vis spectrophotometer. A calibration curve was prepared using gallic acid (0.5–10 mg/L), and the results were expressed in gallic acid equivalents (GAE). All measurements were performed in triplicate.

The DPPH assay was adapted from the microplate method [44]. Extract solutions were prepared in ethanol at different concentrations: 25–250 mg/L for OP and 100–2500 mg/L for PFP. To perform the assay, 20 μL of the sample was added to 180 μL of DPPH solution (150 μmol/L). The microplate was left in the dark at room temperature for 40 min. Afterward, the absorbance was determined at 515 nm. As a control, 20 μL of water was incubated with 180 μL DPPH solution; as blank, 20 μL of water was incubated with 180 μL of ethanol. The percentage of inhibition of DPPH was determined using Equation (1). The IC_50_ (the concentration of extract necessary to inhibit 50% of DPPH) values of each extract were determined.
(1)$$\mathrm{DPPH}\;\mathrm{inhibition}\;\left(\%\right)=\left(1-\frac{{\mathrm{Abs}}_{\mathrm{sample}}-{\mathrm{Abs}}_{\mathrm{blank}}}{{\mathrm{Abs}}_{\mathrm{control}}-{\mathrm{Abs}}_{\mathrm{blank}}}\right)\times 100$$

The ABTS assay was performed according to the literature [45]. Extract solutions were prepared in ethanol at different concentrations: 6.25–62.5 mg/L for OP and 100–1000 mg/L for PFP. To perform the assay, 20 μL of the sample was added to 180 μL of ABTS solution. The microplate was incubated for 15 min at room temperature in the dark. Then, the absorbance was determined at 734 nm. As a control, 20 μL of 0.05 M acetic acid buffer solution (pH 4.6) was incubated with 180 μL ABTS solution. The percentage of inhibition of ABTS was determined using Equation (2). The Trolox equivalent antioxidant capacity (TEAC) of OP and PFP was determined.
(2)$$\mathrm{ABTS}\;\mathrm{inhibition}\;\left(\%\right)=\frac{{\mathrm{Abs}}_{\mathrm{control}}-{\mathrm{Abs}}_{\mathrm{sample}}}{{\mathrm{Abs}}_{\mathrm{control}}}\times 100$$

#### 3.2.4. Antibacterial Activity

The antibacterial activity of the extracts was assessed according to literature protocols [37]. The selected microorganisms were *Escherichia coli*, *Staphylococcus aureus* and *Staphylococcus epidermidis*. Plate count agar (PCA) was the culture medium selected. For OP and PFP, solutions with concentrations of 250 mg/mL and 500 mg/mL in 2% aqueous DMSO were prepared. Ascorbic acid was used as the positive control (antibacterial) and ultrapure water was the negative control. First, bacterial suspensions were prepared in a 0.9% NaCl solution, with an optical density (OD) of 0.1 at 610 nm. Then, PCA plates were incubated with the bacterial suspensions and sterile disks were added to the plates. Afterward, 7 µL aliquots of samples/controls (triplicates) were added to the disks, and the plates were incubated at 37 °C for 24 h. Finally, the diameter of the inhibition halos was measured.

#### 3.2.5. Formulation Production

To evaluate the outcome of the extracts in cosmetic formulations and their effects on the stability and performance of sunscreen, 9 oil-in-water (O/W) moisturizing cream formulations were produced according to the literature with slight modifications [35]. The main difference in the composition of the moisturizers was the exchange between the two emulsifiers: betaine was used as a primary emulsifier, whilst lecithin was selected as a secondary emulsifier. Additives, such as synthetic antioxidant BHT and synthetic UV filter oxybenzone—used as positive controls—and the natural extracts of OP and PFP, were added to the formulations in accordance with Table 6. Notably, the concentrations of both BHT and oxybenzone used are within the legal limits described in the literature. After production, the nine formulations were stored in glass flasks and kept in the dark until further analysis. All the formulations were studied regarding their stability; however, only AOx formulations were evaluated for the antioxidant assays, and only the UV formulations were analyzed for sunscreen protection. The formulations were studied for 28 days at three different analysis times: t_0_ (same week of the production of the samples), t_1_ (second week) and t_2_ (fourth week).

#### 3.2.6. Antioxidant Capacity of Sunscreens

To extract the PCs from the sunscreen formulation, 8 mL of ethanol was added to 2 g of each sunscreen formulation. The solution was homogenized in a vortex for 1 min and placed in an ultrasound bath for 5 min. These steps were repeated three times. Afterward, the solution was centrifuged for 20 min at 3000 rpm. The supernatant was collected and stored in the dark at 4 °C. These solutions were used to determine the radical scavenging activity of the sunscreen formulations, with the same protocols described in Section 3.2.3.

#### 3.2.7. Antibacterial Capacity of Sunscreens

For sunscreens, a similar protocol was followed to the one depicted in Section 3.2.4. However, instead of adding paper disks, small wells were made in the PCA medium with the larger part of glass Pasteur pipettes. The formulations were then pipetted into the wells, in triplicates. The plates were incubated at 37 °C for 24 h, and the diameter of the halos was measured.

#### 3.2.8. Sun Protection Factor

To determine the photoprotection capacity of the OP and PFP extracts and the cream formulations, the sun protection factor (SPF) of these samples was determined [14]. Ethanolic solutions of both extracts and formulations (0.2 mg/mL) were prepared, and the respective absorption values were measured in a UV-3100PC spectrophotometer (VWR, Darmstadt, Germany), between 280 nm and 400 nm with a 1 cm optical path length quartz cell. The SPF values were calculated using Equation (3), where CF is the correction factor equal to 10, EE(λ) refers to the erythemal effect spectrum, I(λ) refers to the solar intensity spectrum for a given wavelength λ, and EE(λ) × I(λ)is the normalized product function. Abs(λ) refers to the absorbance value of the sample at the wavelength λ. For the SPF determination of the extracts, oxybenzone was used as a positive control.
(3)$$\mathrm{SPF}=\mathrm{CF}\;\times \sum_{290}^{320}\mathrm{EE}\left(\lambda \right)\times \;I\left(\lambda \right)\times \;\mathrm{Abs}\left(\lambda \right)$$

#### 3.2.9. Sunscreen Formulation Stability Tests

The stability of the produced sunscreen formulations was analyzed for 4 weeks. Thermal, physical and oxidative stabilities were studied according to the literature, with slight modifications, and the pH value of each formulation was monitored during this period.

##### Determination of Formulation pH

In the analysis of the pH values of all formulations, 9 mL of water was added to 1 g of each sample and the pH was determined under constant agitation using a pH meter.

##### Accelerated Thermal Stability: Temperature Variation Test

All formulations were first incubated at 50 °C overnight (high-temperature phase), followed by a resting phase at room temperature during the subsequent day. Then, the formulations were incubated at 5 °C overnight (low-temperature phase), followed by another resting phase under the previous conditions. Mass, color and consistency changes were visually analyzed to evaluate the thermal stability of the sunscreens.

##### Accelerated Physical Stability: Centrifugation Test

The formulations were centrifuged using a Rotofix 32A (Hettich, Germany) at 4000 rpm for 10 min to evaluate the physical stability. Phase separation and color changes were visually analyzed to evaluate the physical stability of the sunscreens.

##### Oxidative Stability: Peroxide Value

The peroxide value (PV) is an indicator of the primary lipid oxidation of a sample. To determine the PV, 0.1 g of each formulation was dissolved in 9.8 mL of chloroform:methanol (7:3 *v*/*v*). Then, 50 µL of an ammonium thiocyanate solution (30 g/100 mL of water) was added, and the mixture was vortexed for 2–4 s. Afterward, 50 µL of iron (II) solution was added, and the mixture was again vortexed for 2–4 s. Finally, the solution was incubated in the dark at room temperature for 5 min, and the absorbance was measured at 500 nm. PV was calculated using Equation (4).
(4)$$\mathrm{PV}=\frac{\left({\mathrm{Abs}}_{\mathrm{sample}}-{\mathrm{Abs}}_{\mathrm{blank}}\right)\times \;m}{w_{\mathrm{sample}}\times {\;\mathrm{MM}}_{\mathrm{iron}}\times 2}$$
where Abs_sample_ refers to the absorbance of the sample, Abs_blank_ refers to the blank absorbance, m refers to the slope value of the calibration curve of Fe^3+^, w_sample_ corresponds to the sample weight and MM_iron_ corresponds to the molar mass of iron.

#### 3.2.10. Statistical Analysis

To compare the obtained results, a statistical one-way analysis of variance (ANOVA) was performed by calculating the *p*-value (95% confidence), where results with *p*-values less than 0.05 were considered significantly different.

## 4. Conclusions

The primary purpose of the present work was to investigate the feasibility of valorizing agricultural by-products—onion peels and passion fruit peels—by incorporating them into a cosmetic product, specifically a sunscreen. For that, phenolic extracts from both by-products were extracted and characterized. The characterization revealed that the onion peel (OP) extract exhibited higher phenolic content and antioxidant and antibacterial capacity, as well as a higher sun protection factor (SPF). Subsequently, the extracts were incorporated into O/W formulations in different quantities to evaluate their stability, antioxidant capacity and SPF. The stability results revealed that the incorporation of the phenolic extracts did not influence the antibacterial capacity or the thermal and physical stability of the formulations. However, the incorporation of the extracts leads to a slight decrease in the pH values. In terms of lipid oxidation, the formulations containing the extracts had a lower oxidation level than the formulations without additives but had a higher level of oxidation when compared to BHT, a synthetic antioxidant. Regarding the antioxidant capacity, the formulations containing extracts demonstrated analogous capacity in inhibiting DPPH and ABTS, with better results over time, revealing that OP and PFP are valuable natural sources of antioxidants. Lastly, preliminary SPF studies revealed that formulations containing OP and PFP extracts provide significantly less photoprotection than the formulation with oxybenzone (synthetic UV filter). Nevertheless, results illustrated that both extracts (mainly OP) have a photoprotective capacity and may synergistically enhance the effectiveness of oxybenzone in a sunscreen. Overall, the results support the hypothesis that OP and PFP may be valuable sources of phenolic compounds with antioxidant and photoprotective properties. As a result, OP and PFP extracts could be used in various personal care products, including daily moisturizers. Furthermore, the use of phenolic compounds extracted from by-products is critical for complying with sustainability and circular economy principles. Nonetheless, to ensure the safety of the by-products used, further work on the analysis of possible pesticides as well as toxicity assays should be carried out.

## Figures and Tables

**Figure 1 molecules-28-02037-f001:**
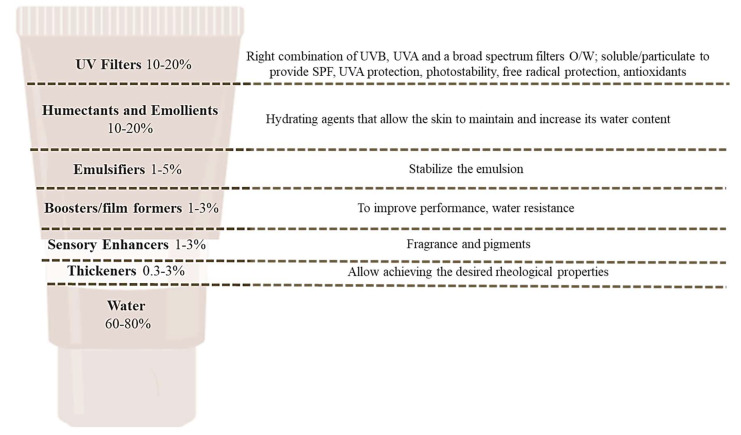
Chemical sunscreen and physical sunscreen, and some examples of the main UV filters used in each case.

**Figure 2 molecules-28-02037-f002:**
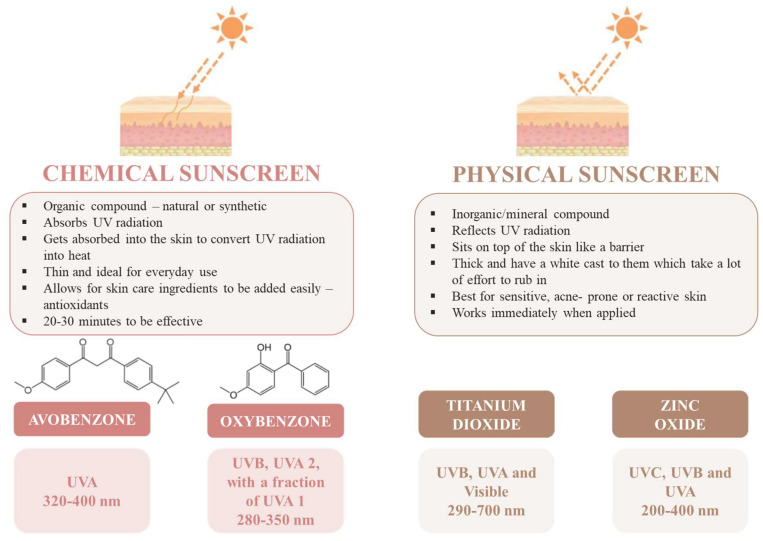
Chemical sunscreen and physical sunscreen, and some examples of the main UV filters used in each case. The wavelength ranges correspond to the light absorption.

**Figure 3 molecules-28-02037-f003:**
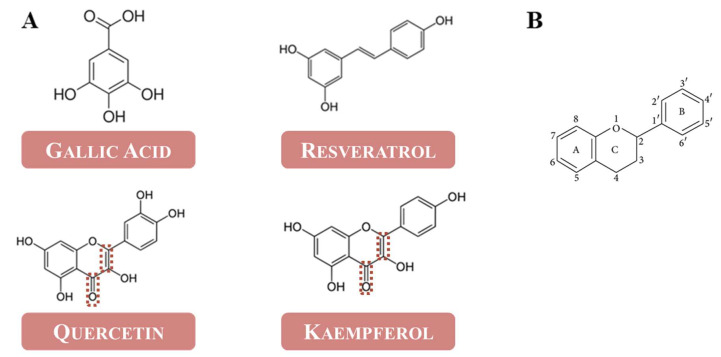
(**A**) Molecular structure of the analyzed phenolics in OP and PFP extract; (**B**) basic structure of flavonoids.

**Figure 4 molecules-28-02037-f004:**
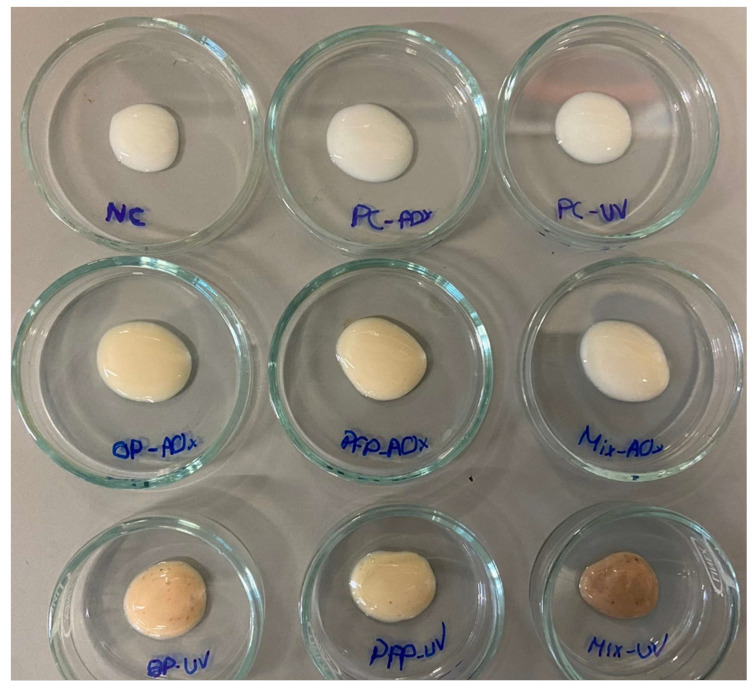
Produced moisturizing creams. NC: negative control; PC-AOx: positive antioxidant control; OP-AOx: onion peel extract antioxidant sample; PFP-AOx: passion fruit peel extract antioxidant sample; Mix-AOx: mixture of antioxidant control and OP and PFP extract; PC-UV: positive UV filter control; OP-UV: onion peel extract UV filter sample; PFP-UV: passion fruit peel extract UV filter sample; Mix-UV: mixture of UV filter control and OP and PFP extract.

**Figure 5 molecules-28-02037-f005:**
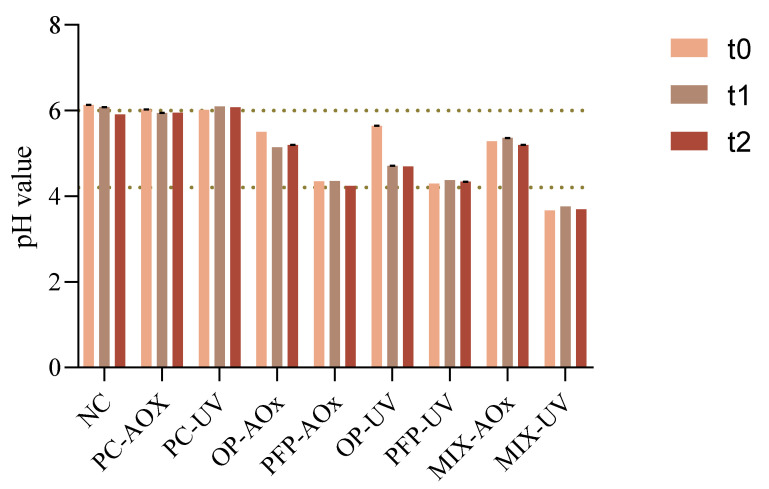
Variation in the pH value of the different formulations for the three study times: t_0_—week of production; t_1_—two weeks after production; t_2_—four weeks after production. NC: negative control; PC-AOx: positive antioxidant control; OP-AOx: onion peel extract antioxidant sample; PFP-AOx: passion fruit peel extract antioxidant sample; Mix-AOx: mixture of antioxidant control and OP and PFP extract; PC-UV: positive UV filter control; OP-UV: onion peel extract UV filter sample; PFP-UV: passion fruit peel extract UV filter sample; Mix-UV: mixture of UV filter control and OP and PFP extract.

**Figure 6 molecules-28-02037-f006:**
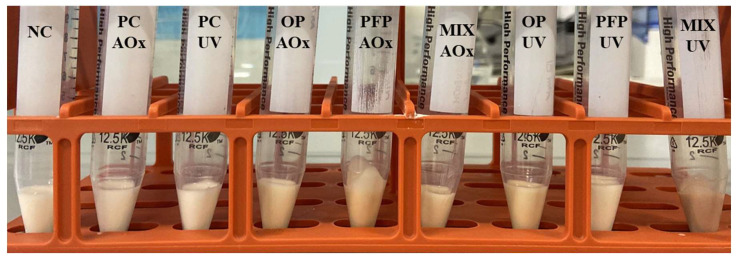
Results obtained for the centrifuge test. NC: negative control; PC-AOx: positive antioxidant control; OP-AOx: onion peel extract antioxidant sample; PFP-AOx: passion fruit peel extract antioxidant sample; Mix-AOx: mixture of antioxidant control and OP and PFP extract; PC-UV: positive UV filter control; OP-UV: onion peel extract UV filter sample; PFP-UV: passion fruit peel extract UV filter sample; Mix-UV: mixture of UV filter control and OP and PFP extract.

**Figure 7 molecules-28-02037-f007:**
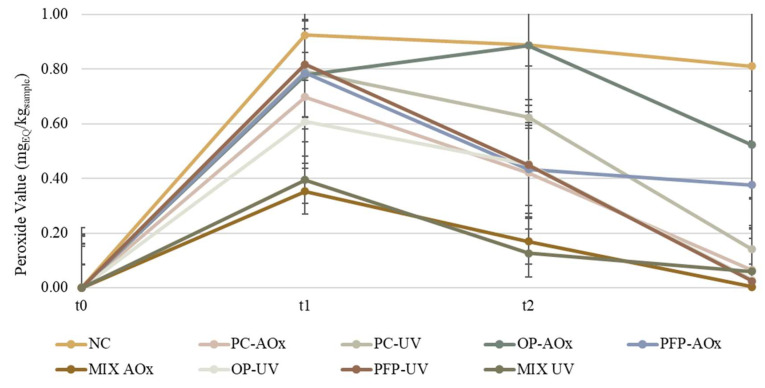
Variation in the peroxide value throughout the study period for the produced formulations. The analysis was performed for three study times: t_0_—week of production; t_1_—two weeks after production; t_2_—four weeks after production. NC: negative control; PC-AOx: positive antioxidant control; OP-AOx: onion peel extract antioxidant sample; PFP-AOx: passion fruit peel extract antioxidant sample; Mix-AOx: mixture of antioxidant control and OP and PFP extract; PC-UV: positive UV filter control; OP-UV: onion peel extract UV filter sample; PFP-UV: passion fruit peel extract UV filter sample; Mix-UV: mixture of UV filter control and OP and PFP extract.

**Figure 8 molecules-28-02037-f008:**
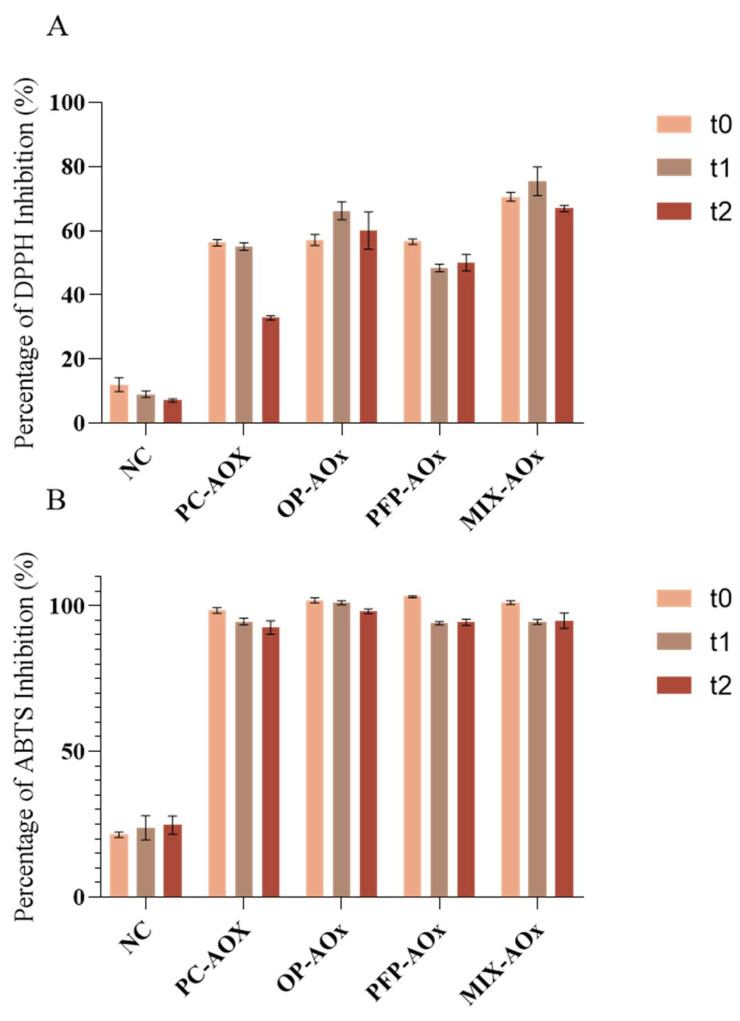
Results were obtained for (**A**) assay with DPPH radical and (**B**) assay with ABTS radical. The analysis was performed for three study times: t_0_—week of production; t_1_—two weeks after production; t_2_—four weeks after production. NC: negative control; PC-AOx: positive antioxidant control; OP-AOx: onion peel extract antioxidant sample; PFP-AOx: passion fruit peel extract antioxidant sample; Mix-AOx: mixture of antioxidant control and OP and PFP extract.

**Figure 9 molecules-28-02037-f009:**
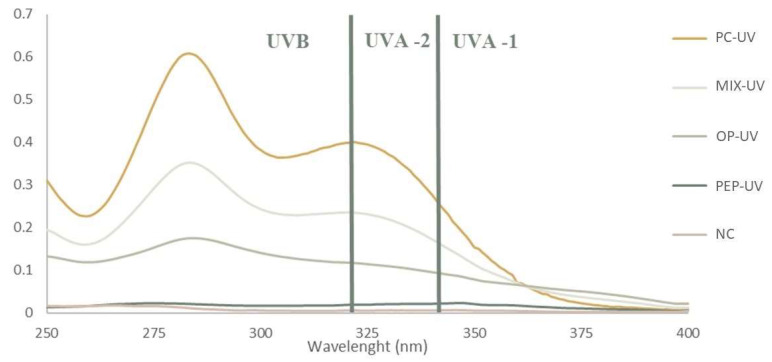
Absorption spectra used to determine the sun protective factor (SPF) of formulations. NC: negative control; PC-UV: positive UV filter control; OP-UV: onion peel extract UV filter sample; PFP-UV: passion fruit peel extract UV filter sample; Mix-UV: mixture of UV filter control and OP and PFP extract.

**Table 1 molecules-28-02037-t001:** Obtained results in the characterization of the biological capacity of the onion peel and passion fruit peel extracts.

			OP	PFP
Antibacterial Capacity (d_halo_, mm)	C = 500 mg/mL	*E. coli*	N. D.	N. D.
		*S. aureus*	16.7 ± 0.5 ^a^	10.3 ± 0.9 ^b^
		*S. epidermidis*	18.3 ± 0.5 ^a^	8.5 ± 0.5 ^b^
	C = 250 mg/mL	*E. coli*	N. D.	N. D.
		*S. aureus*	7.0 ± 1.4	N. D.
		*S. epidermidis*	11.7 ± 0.8	N. D.
TPC (mg_GAE_/g)			377.4 ± 16.4 ^a^	124.2 ± 9.3 ^b^
DPPH		IC_50_ (mg/mL)	15.9 ± 0.4 ^a^	112.2 ± 2.5 ^b^
		TE (mg_TE_/g)	308.9 ± 2.8 ^a^	38.5 ± 3.1 ^b^
ABTS (TEAC) (mg_TE_/g)			413.4 ± 22.5 ^a^	39.0 ± 1.4 ^b^

The results are stated as mean ± standard deviations of 3 independent measurements. Different letters (a–b), in the same line, represent statistically different values (*p* < 0.05) between the OP and PFP extracts. OP: onion peel; PFP: passion fruit peel; N. D.: not detected; TPC: total phenolic content; GAE: gallic acid equivalents; IC_50_: the concentration of extract necessary to inhibit 50% of DPPH radical; TE: Trolox equivalents; TEAC: Trolox equivalent antioxidant capacity.

**Table 2 molecules-28-02037-t002:** HPLC/DAD analysis results: calibration curve, coefficient of determination (R^2^), linearity range and the concentration of each compound in both onion peel extract and passion fruit peel extract.

Compound	Calibration Curve	R^2^	Linearity	Concentration (mg/g_extract_)	
				OP	PFP
Gallic Acid	A = 1.21 × 10^5^ C + 1.33 × 10^6^	0.9978	100	1.33 ± 0.50 ^a^	1.46 ± 0.18 ^a^
Kaempferol	A = 7.34 × 10^5^ C + 2.41 × 10^5^	0.9993	80	1.43 ± 0.05 ^a^	0.62 ± 0.07 ^b^
Quercetin	A = 7.37 × 10^5^ C − 2.68 × 10^5^	0.9994	80	26.44 ± 1.02 ^a^	2.08 ± 0.09 ^b^
Resveratrol	A = 1.42 × 10^6^ C + 2.79 × 10^5^	0.9999	80	4.79 ± 0.70 ^a^	0.28 ± 0.01 ^b^

The results are expressed as mean ± standard deviations of 3 independent measurements. Different letters (a–b) in the same line represent statistically different values (*p* < 0.05) between the OP and PFP extracts. OP: onion peel; PFP: passion fruit peel; A: peak area; C: compound concentration (mg/L).

**Table 3 molecules-28-02037-t003:** Sun protection factor (SPF) for oxybenzone (synthetic UV filter), onion peel extract and passion fruit peel extract.

Sample	Concentration (mg/mL)	Abs for the Main Peaks		SPF
Oxybenzone	0.01	0.743	0.489	3.3 ± 0.5 ^a^
OP	0.2	0.608	0.401	8.8 ± 0.3 ^b^
PFP	0.2	0.351	0.253	3.3 ± 0.7 ^a^

Different letters (a–b) in the same column represent statistically different values (*p* < 0.05) between oxybenzone and the OP and PFP extracts. OP: onion peel; PFP: passion fruit peel.

**Table 4 molecules-28-02037-t004:** Antibacterial capacity, represented through the diameter of the inhibition halo in mm, of the formulations against *S. aureus*, *S. epidermidis* and *E. coli* for t_0_ and t_2_.

Sample	*S. aureus*		*S. epidermidis*		*E. coli*	
	t_0_	t_2_	t_0_	t_2_	t_0_	t_2_
NC	12.7 ± 0.5 ^a,A^	12.3 ± 3.3 ^a,A^	11.0 ± 1.4 ^a,A^	13.0 ± 1.6 ^a,b,A^	<5.0	<5.0
PC-AOx	11.7 ± 1.7 ^a,A^	10.7 ± 1.7 ^a,A^	15.0 ± 0.0 ^b,c,A^	11.0 ± 0.0 ^a,B^	<5.0	<5.0
PC-UV	14.3 ± 0.5 ^a,A^	11.7 ± 0.9 ^a,A^	15.3 ± 1.2 ^d,A^	15.0 ± 0.8 ^a,A^	<5.0	<5.0
OP-AOx	11.0 ± 0.0 ^a,A^	9.3 ± 0.9 ^a,A^	12.0 ± 0.8 ^a,b,A^	11.7 ± 0.5 ^a,A^	<5.0	<5.0
PFP-AOx	11.3 ± 1.2 ^a,A^	11.0 ± 1.6 ^a,A^	13.7 ± 0.5 ^a,A^	11.0 ± 0.5 ^a,B^	<5.0	<5.0
MIX-AOx	11.0 ±1.6 ^a,A^	8.7 ± 0.5 ^a,A^	15.7 ± 2.1 ^c,A^	11.7 ± 1.9 ^a,B^	<5.0	<5.0
OP-UV	10.0 ± 0.8 ^a,A^	9.7 ± 1.2 ^a,A^	13.7 ± 0.9 ^a,d,A^	13.7 ± 0.9 ^b,A^	<5.0	<5.0
PFP-UV	10.0 ± 0.8 ^a,A^	8.7 ± 0.5 ^a,A^	12.0 ± 0.8 ^a,d,A^	11.3 ± 0.5 ^b,A^	<5.0	<5.0
MIX-UV	13.0 ± 1.6 ^a,A^	12.0 ± 0.8 ^a,A^	12.7 ± 0.5 ^a,d,A^	15.5 ± 0.4 ^a,A^	<5.0	<5.0

The results are expressed as mean ± standard deviations of 3 independent measurements. Different lowercase letters (a–d) in the same column represent statistically different values (*p* < 0.05) between the formulations for each analysis time. Different capital letters (A–B) in the same line, and for the same bacteria, represent statistically different values (*p* < 0.05) between the two analysis times for each formulation. NC: negative control; PC-AOx: positive antioxidant control; OP-AOx: onion peel extract antioxidant sample; PFP-AOx: passion fruit peel extract antioxidant sample; Mix-AOx: mixture of antioxidant control and OP and PFP extract; PC-UV: positive UV filter control; OP-UV: onion peel extract UV filter sample; PFP-UV: passion fruit peel extract UV filter sample; Mix-UV: mixture of UV filter control and OP and PP extract.

**Table 5 molecules-28-02037-t005:** SPF values for the formulations and absorbance values for the main peaks.

Sample	Abs for the Main Peaks		SPF
NC	0.018	0.016	0.34
PC-UV	0.608	0.401	10.10
OP-UV	0.175	0.144	3.19
PFP-UV	0.023	0.021	0.44
MIX-UV	0.351	0.253	6.05

NC: negative control; PC-UV: positive UV filter control; OP-UV: onion peel extract UV filter sample; PFP-UV: passion fruit peel extract UV filter sample; Mix-UV: mixture of UV filter control and OP and PFP extract.

**Table 6 molecules-28-02037-t006:** Percentage of additives added to the nine formulations produced.

Sample	BHT	Oxybenzone	OP	PFP
	(%)			
NC	-			
PC-AOx	0.5	-	-	-
OP-AOx	-	-	0.5	-
PFP-AOx	-	-	-	0.5
MIX-AOx	0.2	-	0.2	0.2
PC-UV	-	5.0	-	-
OP-UV	-	-	5.0	-
PFP-UV	-	-	-	5.0
MIX-UV	-	1.6	1.6	1.6

NC: negative control; PC-AOx: positive antioxidant control; OP-AOx: onion peel extract antioxidant sample; PFP-AOx: passion fruit peel extract antioxidant sample; Mix-AOx: mixture of antioxidant control and OP and PFP extract; PC-UV: positive UV filter control; OP-UV: onion peel extract UV filter sample; PFP-UV: passion fruit peel extract UV filter sample; Mix-UV: mixture of UV filter control and OP and PFP extract.

## Data Availability

Not applicable.

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
