# Peer review of "A Novel Approach in Skin Care: By-Product Extracts as Natural UV Filters and an Alternative to Synthetic Ones"

_molecules, 2023, doi:10.3390/molecules28052037_

Round 1
Reviewer 1 Report
Dear authors, the comments are attached.
Best regards.

Author Response
The topic is very interesting and relevant, taking into account the use of agro-industrial by-products and the valorization of its bioactive composition. I believe that this point should be given greater prominence in the text.
Answer: Thank you for your comment. The authors agree and added the suggested information.
I missed, in the introduction, a call to attention to the novelty of the study. Are there already studies that address this topic? Which plant extracts have been studied for this purpose? What are the main results reported in the literature?
Answer: Thank you for your comment. There are few studies regarding the potential of plant and fruit extracts to be used as UV filters in sunscreen. To the authors’ best knowledge, this is the first study analyzing the potential use of agricultural by-products (onion peel and passionfruit peel) as UV filters in sunscreens, studying the effect of their incorporation on the stability of the formulation.
The authors could insert a brief justification for the choice of microorganisms Escherichia coli, Staphylococcus aureus and Staphylococcus epidermidis.
Answer: Thank you for your comment. The authors agree and added a brief justification to the manuscript as suggested.
“All the formulations were studied regarding their stability; however, only AOx formulations were evaluated for the antioxidant assays, and for the sunscreen protection only the UV formulations were analyzed”. Could the authors describe why?
Answer: In this work, two positive controls were used. One with BHT, a synthetic antioxidant, which was used as a positive control for the antioxidant assays and one with oxybenzone, a synthetic UV filter, which was used as a positive control for the sunscreen protection analysis. Since the legal limits of these two compounds (BHT and oxybenzone) are different, the authors opted to create AOX formulations and UV formulations, that differ in the number of additives added, following the legal limits for BHT and oxybenzone, respectively. Therefore, only AOX formulations were evaluated in the antioxidant assays since they present the same amount of additives as the positive control with BHT to allow a direct comparison between the extracts and the synthetic antioxidant. The same for the UV filter analysis.
Was it thought of some kind of sensory analysis? Did the addition of onion skin not interfere with the aroma of the product? This could be a limiting factor for commercialization and consumer acceptance.
Answer: Thank you for your comment. The authors understand the importance of performing a sensory analysis of the product. However, the primary purpose of this work was to understand the potential properties of the extracts studied that can be advantageous for their incorporation on sunscreens. The authors decided that it was better to perform a sensory analysis in future work when the sunscreen formulations are optimized.
Additionally,
- “Ethanol was used as a solvent, with a ratio of 1:20 m/V.”?
Answer: In the extraction process, ethanol was the solvent used. The ratio of mass of sample to volume of solvent chosen was 1:20 (m/V).
- Figures 1 and 2 are low resolution, I encourage the authors to improve the images format.
Answer: Thank you for your comment. The resolution of Figures 1 and 2 was improved.
- Abbreviations should not be indicated in the abstract. It is better to use the full form of writing.
Answer: Thank you for your comment. The authors are aware that abbreviations should be avoided in the abstract; however, due to the word limit in that section, abbreviations had to be used.
- Scientific names must be written in italics.
Answer: Thank you for your comment. The text was corrected.
- The introduction can be better substantiated based on scientific papers available in the literature.
- Equations must be numbered and cited in the text.
Answer: Thank you for your comment. The text was corrected.
- Figures must be numbered and cited in the text.
Answer: Thank you for your comment. The text was corrected.
- In general, I believe that the abstract needs contextualization, the introduction can be better grounded, the methodology must be detailed, and the conclusions need a critical point of view.
Answer: Thank you for your comment. Some alterations were made in the text, as suggested.

Reviewer 2 Report
The authors investigated the antioxidant and photoprotective properties of onion peel and passion fruit peel extracts in topical formulations as an alternative to synthetic antioxidants and UV filters. The authors suggest that OP and PFP can be incorporated in daily moisturizers with SPF and sunscreens replacing and/or diminishing the quantities of synthetic ingredients, reducing their negative effects on human health and the environment.
Comments:
-The authors stated that inorganic filters, such as zinc oxide and titanium dioxide, as well as some other synthetic ingredients in sunscreens, negatively affect human health and can result in hazards. The authors should cite recent studies on humans regarding this matter and reference appropriate sources for these claims.
-What studies have authors performed or plan to perform to prove that OP and PFP have no adverse side effects on human health? The safety of compounds intended to be used in humans should be assessed regardless of whether the compound is of natural or synthetic origin. For example, it is known that quercetin and resveratrol can exhibit cytotoxic effects in a dose-dependent manner. Therefore, the authors should discuss this topic, keeping in mind the polyphenolic and other content of the OP and PFP.
-The authors should discuss the photostability of OP and PHP ingredients and the eventual negative effects on human health of their photodegradation products (upon UV irradiation).
- Tables 1-3. The results of the statistical analysis are not clearly presented. For example, what comparisons are represented by the letters a and b? Figure legends should be more informative.
- It looks like SPFs for investigated compounds are very low, especially of PFP (similar to negative control in formulation). Why did the authors consider PFP a promising alternative to a UV filter in sunscreens? In general, according to this study, the PFP exhibited less favorable characteristics when compared to OP.
-Table 4. The results of the statistical analysis are not clearly presented. It is unclear whether data for the OP and PFP were compared with each other or with controls. What comparisons are represented by all the letters?
- The authors used different concentrations of OP and PFP for the various experiments and made the conclusions of their properties based on them. How does it affect the findings of the study? (e.g., concentrations of OP and PFP used for the antibacterial activity assessment were much higher than those used in other experiments). What concentrations of OP and PFP should be present in the sunscreens for their optimal functioning?
-Figures 5, 7, and 8. Figure legends should provide information regarding the study times length.
Author Response
Comments and Suggestions for Authors
The authors investigated the antioxidant and photoprotective properties of onion peel and passion fruit peel extracts in topical formulations as an alternative to synthetic antioxidants and UV filters. The authors suggest that OP and PFP can be incorporated in daily moisturizers with SPF and sunscreens replacing and/or diminishing the quantities of synthetic ingredients, reducing their negative effects on human health and the environment.
Comments:
The authors stated that inorganic filters, such as zinc oxide and titanium dioxide, as well as some other synthetic ingredients in sunscreens, negatively affect human health and can result in hazards. The authors should cite recent studies on humans regarding this matter and reference appropriate sources for these claims.
Answer: Thank you for your comment. The authors added more references regarding the potential negative effects of inorganic filters and synthetic ingredients in sunscreens.
What studies have authors performed or plan to perform to prove that OP and PFP have no adverse side effects on human health? The safety of compounds intended to be used in humans should be assessed regardless of whether the compound is of natural or synthetic origin. For example, it is known that quercetin and resveratrol can exhibit cytotoxic effects in a dose-dependent manner. Therefore, the authors should discuss this topic, keeping in mind the polyphenolic and other content of the OP and PFP.
Answer: Thank you for your comment. Although not included in the scope of the work developed, the authors understand the importance of studying the adverse effects of OP and PFP on human health. Therefore, it would be of great interest to evaluate the toxicity of the OP and PFP extracts and the produced sunscreens in further work. The cytotoxicity of the extracts could be evaluated in keratinocytes, the most common cell type in the skin since the final product is intended for topical applications. For that, the cell membrane permeability, i.e. the viability of the cells, and the metabolic activity of the cells could be analyzed by, for example, trypan blue exclusion assay and AlamarBlue reduction assay, respectively. These tests could give us an indication of which concentrations of OP and PFP extracts are safe for skin cells. Afterwards, skin sensitization could be evaluated with formulations containing safe concentrations of the extracts to see if any allergic response occurs in susceptible individuals. Finally, since the intended application is sunscreens, which are exposed to sunlight, photo-induced toxicity should also be evaluated. However, these studies are outside the scope of this article.
The authors should discuss the photostability of OP and PHP ingredients and the eventual negative effects on human health of their photodegradation products (upon UV irradiation).
Answer: Thank you for your comment. The authors did not find anything in the literature related to the photostability of the OP and PFP ingredients. It would be of great interest to study this topic in future work to ensure the safety of the use of OP and PFP extracts in sunscreen formulations.
Tables 1-3. The results of the statistical analysis are not clearly presented. For example, what comparisons are represented by the letters a and b? Figure legends should be more informative.
Answer: Thank you for your comment. The suggested alterations were made in the legends of Tables 1, 2 and 3.
It looks like SPFs for investigated compounds are very low, especially of PFP (similar to negative control in formulation). Why did the authors consider PFP a promising alternative to a UV filter in sunscreens? In general, according to this study, the PFP exhibited less favorable characteristics when compared to OP.
Answer: Thank you for your comment. Although the SPF obtained for PFP was low, the formulations containing the PFP extract presented other advantageous biological activities, such as antioxidant capacity. The authors understand that PFP extract, by itself, can not replace synthetic UV filters. However, and since they display other interesting properties, PFP extracts can be used in combination with synthetic UV filters or other natural compounds with more promising UV filter capacity, such as the OP extract, in sunscreens formulations. Additionally, the concentration of PFP extract present in the sunscreen could be increased, since the legal limits for UV filters are related to one compound, whereas PFP extract contains several different compounds with different characteristics.
Table 4. The results of the statistical analysis are not clearly presented. It is unclear whether data for the OP and PFP were compared with each other or with controls. What comparisons are represented by all the letters?
Answer: Thank you for your comment. The suggested alterations were made in the legends of Table 4. Data from OP and PFC were compared with each other and with the respective controls (NC and PC-AOX for AOX formulations and NC and PC-UV for UV formulations).
The authors used different concentrations of OP and PFP for the various experiments and made the conclusions of their properties based on them. How does it affect the findings of the study? (e.g., concentrations of OP and PFP used for the antibacterial activity assessment were much higher than those used in other experiments). What concentrations of OP and PFP should be present in the sunscreens for their optimal functioning?
Answer: Thank you for your comment. The selected concentrations for the antibacterial analysis of the extracts were based on the concentrations used in the literature. The authors consider that the different concentrations used do not affect the findings, since the biological assays performed are used to determine different characteristics of the extracts and, therefore, the concentration should be adjusted to each test. Regarding the MIX formulation, the percentage of each extract was 1.6%; the authors believe that this percentage can be increased since the extracts are a mixture of compounds and contain additional beneficial chemical compounds, such as vitamins. However, in order to determine the optimal, the UV spectrum for different concentrations of the extracts should be obtained. Additionally, more formulations could be assessed, for example, a formulation containing 10% of extract and a mix formulation containing 1.6% of oxybenzone and 5% of each extract. Nevertheless, the intended goal of this study was to prove that there is potential for some phenolic extracts to replace and/or decrease the concentration of synthetic UV filters. The authors believe that the determination of the optimal concentrations of the extract should be performed as future work, as well as some other studies, for example, a sensorial analysis.
Figures 5, 7, and 8. Figure legends should provide information regarding the study times length.
Answer: Thank you for your comment. The suggested alterations were made in the legends of Figures 5, 7 and 8.

Reviewer 3 Report
Dear authors,
The manuscript (molecules-2210381) has been reviewed.
The main question addressed by the research is to investigate and assess the antioxidant and photoprotective properties of onion peel (OP) and passion fruit peel (PFP) extracts in topical formulations as an alternative to synthetic antioxidants and UV filters.
The topic is original and relevant in the field. It addresses a novelty with the investigation of the feasibility of valuing agricultural by-products – onion peels and passion fruit peels by incorporating them into a cosmetic product, specifically a sunscreen.
The methodology is clear and well presented
The results are well presented and discussed
The conclusions consistent with the evidence and arguments presented
and they address the main question.
The references appropriate, mostly new and support the discussion of the findings.
However, some editing is needed as:
- In Keywords section, the word sustainability should be removed from the list and replaced by an other term related to the findings of this research paper.
- The conclusion is too long and should be summarized and reduced
- In figures2, 5 and 8, the colors used for text or histograms should be changed and be different.
Author Response
Dear authors,
The manuscript (molecules-2210381) has been reviewed.
The main question addressed by the research is to investigate and assess the antioxidant and photoprotective properties of onion peel (OP) and passion fruit peel (PFP) extracts in topical formulations as an alternative to synthetic antioxidants and UV filters.
The topic is original and relevant in the field. It addresses a novelty with the investigation of the feasibility of valuing agricultural by-products – onion peels and passion fruit peels by incorporating them into a cosmetic product, specifically a sunscreen.
The methodology is clear and well presented
The results are well presented and discussed
The conclusions consistent with the evidence and arguments presented and they address the main question.
The references appropriate, mostly new and support the discussion of the findings.
Answer: The authors thank you for the comments above.
However, some editing is needed as:
- In Keywords section, the word sustainability should be removed from the list and replaced by an other term related to the findings of this research paper.
Answer: Thank you for your comment. The authors replaced the word sustainability with another in Keywords (sun protection factor), as suggested.
- The conclusion is too long and should be summarized and reduced
Answer: The conclusion was summarized, as suggested.
- In figures2, 5 and 8, the colors used for text or histograms should be changed and be different.
Answer: Thank you for your comment. The suggested alterations were made.

Round 2
Reviewer 2 Report
-The authors have improved the manuscript on some points. However, important issues regarding the safety of compounds intended for human use were not discussed nor even mentioned in the manuscript. The authors did not include a paragraph explaining that future studies are mandatory to prove the safety and efficacy of OP and PFP, especially regarding their toxicity. The authors did not discuss the photostability of OP and PHP ingredients and the eventual adverse effects on the human health of their photodegradation products (upon UV irradiation). Although it is important to promote the reuse of agro-industrial by-products to protect the environment, the promotion of any compound to be used in humans should include all relevant information. The authors did not emphasize the importance of performing future studies regarding all these issues, which may lead to misconclusions regarding the safety and efficacy of the compounds they investigated.
- The results of the statistical analysis are still confusingly explained. The authors should state a- comparison between X and Y; b- comparison between…, etc.
Author Response
Author's Reply to the Review Report (Reviewer2)
Comments and Suggestions for Authors
The authors have improved the manuscript on some points. However, important issues regarding the safety of compounds intended for human use were not discussed nor even mentioned in the manuscript. Although it is important to promote the reuse of agro-industrial by-products to protect the environment, the promotion of any compound to be used in humans should include all relevant information. The authors did not include a paragraph explaining that future studies are mandatory to prove the safety and efficacy of OP and PFP, especially regarding their toxicity. The authors did not emphasize the importance of performing future studies regarding all these issues, which may lead to misconclusions regarding the safety and efficacy of the compounds they investigated.
Answer: Thank you for your comment. The authors did mention the importance of evaluating the safety of the by-products used, mainly the presence of pesticides. However, information about the safety of the extracts was not present. The authors added that information.
The authors did not discuss the photostability of OP and PHP ingredients and the eventual adverse effects on the human health of their photodegradation products (upon UV irradiation).
Answer: Thank you for your comment. The authors did not find anything in the literature related to the photostability of the OP and PFP ingredients. However, that information was added to the manuscript.
The results of the statistical analysis are still confusingly explained. The authors should state a- comparison between X and Y; b- comparison between…, etc.
Answer: Thank you for your comment. The authors added more information in order to clarify the statistical analysis.